# Highly-Dispersed Ni-NiO Nanoparticles Anchored on an SiO$_2$ Support for an Enhanced CO Methanation Performance

**Jiangwei Li** [1,†], **Panpan Li** [1,†], **Jiangbing Li** [1], **Zhiqun Tian** [2] **and Feng Yu** [1,*] 

1   Key Laboratory for Green Processing of Chemical Engineering of Xinjiang Bingtuan, School of Chemistry and Chemical Engineering, Shihezi University, Shihezi 832003, China; jwlyshz@163.com (J.L.); lppbjjt@163.com (P.L.); ljbin@shzu.edu.cn (J.L.)
2   Collaborative Innovation Center of Sustainable Energy Materials, Guangxi University, Nanning 530004, China; tianzhiqun@gxu.edu.cn
*   Correspondence: yufeng05@mails.ucas.ac.cn; Tel.: +86-993-205-7272; Fax: +86-993-205-7270
†   These authors contributed equally to this work.

**Abstract:** Highly-dispersed Ni-NiO nanoparticles was successfully anchored on an SiO$_2$ support via a one-pot synthesis and used as heterogeneous catalysts for CO methanation. The as-obtained Ni-NiO/SiO$_2$ catalyst possessed a high Ni content of 87.8 wt.% and exhibited a large specific surface area of 71 m$^2$g$^{-1}$ with a main pore diameter of 16.7 nm. Compared with an H$_2$-reduced Ni-NiO/SiO$_2$ (i.e., Ni/SiO$_2$) catalyst, the Ni-NiO/SiO$_2$ displayed a superior CO methanation performance. At the temperature of 350 °C, the Ni-NiO/SiO$_2$ showed a CO conversion of 97.1% and CH$_4$ selectivity of 81.9%, which are much better values than those of Ni/SiO$_2$. After a 50-h stability test, the Ni-NiO/SiO$_2$ catalyst still had an overwhelming stability retention of 97.2%, which was superior to the 72.8% value of the Ni/SiO$_2$ catalyst.

**Keywords:** Ni-NiO nanoparticles; nickel-based catalyst; CO methanation; syngas; synthetic natural gas

---

## 1. Introduction

Fossil fuels and their emissions cause environmental pollution. Therefore, the development of environmentally friendly and sustainable energy is urgently required. The main component of natural gas is methane (CH$_4$), which is characterized by cleanliness, a high calorific value, and a high energy efficiency. This is why it is considered to be an ideal energy source [1,2]. In recent years, the cost of natural gas has increased because of an increase in its demand, and many countries have taken a series of measures to deal with its tight supply. Other than this, synthetic natural gas (SNG) production by methanation from coal or biomass synthesis gas is an effective method. In particular, the production of synthetic gas by CO or CO$_2$ methanation has been extensively studied [3–5].

The active components of methanation catalysts are mainly distributed throughout the group VIIIB [6,7]. Among these metals, Ni-based catalysts exhibit excellent catalytic performances and better CH$_4$ selectivity, which make them usable in the industry [8,9]. CO methanation is performed on a large scale and Ni-based catalysts are the most commonly used [10]. Lu et al. [11] and Wang et al. [12] tested and compared the performances of methanation catalysts containing various metal components and found that Ni-based catalysts have a good performance. Lakshmanan et al. [13] prepared Ni/SiO$_2$ catalysts with different loadings and 55 wt.% Ni@SiO$_2$ catalysts for CO methanation. It was found that the core-shell Ni@SiO$_2$ catalysts had a better performance. Yan et al. [14] prepared an Ni/SiO$_2$ catalyst and applied it to CO methanation, which exhibited superior activity and stability. Zhao et al. [15] prepared an Ni/SiO$_2$ catalyst by different solution impregnation methods for CO methanation research.

The results showed that ammonia impregnation effectively improved the activity and enhanced the high-temperature stability of the Ni/SiO$_2$ catalyst.

It was reported that the morphology distribution and particle size of the CO methanation catalyst significantly affect the catalytic performance, so the catalysts with better dispersion and smaller particles exhibit significantly improved performances [16–18]. Some researchers have prepared Ni-based catalysts with higher dispersion and smaller particles for CO methanation; these measures effectively prevent the sintering of active components and carbon deposition to improve the performance of the catalysts [19].

Here, we report a high Ni content and high-dispersion nickel-based catalysts of Ni-NiO/SiO$_2$ and Ni/SiO$_2$ by a one-pot method for CO methanation. The Ni-Ni/SiO$_2$ catalysts had a metal dispersion that was 0.16% higher than the Ni/SiO$_2$ catalysts of 0.05% and a specific surface area of 71 m$^2$g$^{-1}$ that was superior to the latter's area of 25 m$^2$g$^{-1}$. The two catalysts were subjected to a CO methanation performance test and a 50-h stability test at 350 °C. It was noteworthy that the catalyst of Ni-NiO/SiO$_2$ exhibited exceptional stability and carbon deposition resistance, which may be due to its high dispersion. The high dispersibility of Ni-NiO catalysts increases their activity, and higher Ni loading provides more active sites [20].

## 2. Results and Discussion

Figure 1 shows the X-ray diffraction (XRD) pattern of the fresh catalyst before the reaction. As observed in the figure, the Ni-NiO/SiO$_2$ catalyst exhibited NiO diffraction peaks at 2θ = 37.2° and 62.9° (PDF-# NO. 47-1049). These peaks, which correspond to the (111) and (220) crystal faces of NiO, were relatively wide, indicating that it had a poor crystallinity. The Ni-NiO/SiO$_2$ catalyst exhibited an Ni diffraction peak (2θ = 44.5°, PDF- # NO. 04-0850). The diffraction peaks of the Ni/SiO$_2$ catalyst (2θ = 44.5°, 51.8°, and 76.4°, PDF-# NO. 04-0850) correspond to the (111), (200), and (220) crystal faces of Ni [21]. The diffraction peaks of the NiO/SiO$_2$ catalyst (2θ = 37.2°, 43.3°, 62.9°, 75.4°, and 79.4°, PDF-# NO. 47-1049) correspond to the (111), (200), (220), (311), and (222) crystal faces of NiO, respectively. The Ni diffraction peak for the Ni/SiO$_2$ catalyst was sharp, indicating that it had a high crystallinity. Additionally, no other impurity peaks were observed, which shows that it had a higher purity. Because the uncalcined Ni-NiO/SiO$_2$ catalyst was amorphous and had a poor crystallinity, the NiO and Ni diffraction peaks were relatively wide and offset compared with those of the NiO/SiO$_2$ catalyst and Ni/SiO$_2$ catalyst. NiO(200) and Ni(111) planes were chosen to calculate the mean size of NiO and Ni crystallites by Scherrer's equation [22]. The crystal size of NiO particles was 16.5 nm more than the catalysts NiO/SiO$_2$ and the crystal size of Ni was 14.9 nm for the catalysts Ni/SiO$_2$.

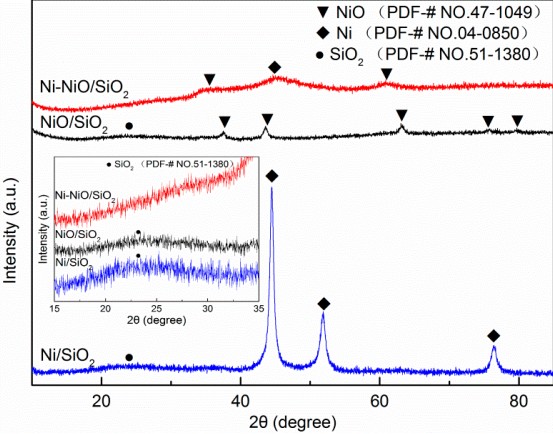

**Figure 1.** XRD patterns of as-obtained Ni-NiO/SiO$_2$, NiO/SiO$_2$, and Ni/SiO$_2$ catalysts (The inner picture is the diffraction peak of SiO$_2$).

To study the oxide state and surface elemental composition, X-ray photoelectron spectroscopy (XPS) analysis of the Ni-NiO/SiO$_2$ and Ni/SiO$_2$ catalysts was performed, and the results are shown in Figure 2. Simultaneously, the peak value was fitted by a Gaussian component to determine the valence states of Ni, O, and Si in the sample. Figure 2a shows the survey peaks of the respective elements in the catalysts Ni-NiO/SiO$_2$ and Ni/SiO$_2$, and it is clear that there was almost no difference between the two catalysts. Figure 2b shows the valence state of Ni in the two catalysts. The typical peaks of the Ni 2p orbit range from 850 eV to 884 eV; both of the catalysts contain Ni 2p3/2 and Ni 2p1/2 orbits. The Ni 2p3/2 orbit can be divided into three small peaks: for catalysts Ni-NiO/SiO$_2$, the three peaks were Ni$^{2+}$ at 856.6 eV, and satellite peaks of Ni 2p3/2 at 858.1 eV and 862.7 eV, respectively. There are three peaks at 856.6 eV, 858.9 eV, and 862.7 eV for Ni/SiO$_2$ catalysts. The Ni 2p1/2 orbit contains two characteristic peaks of Ni$^{2+}$ at 874.1 eV and a satellite peak at 880.2 eV for both catalysts. The catalysts of Ni/SiO$_2$ contain the characteristic Ni$^0$ peak at 853.1 eV, while no peak of Ni$^0$ was observed in the Ni-NiO/SiO$_2$ catalysts, a possible reason for which may be that the Ni in the Ni-NiO/SiO$_2$ catalysts was relatively dispersed and could not be detected [9,23]. Figure 2c shows the characteristic O 1s peak for the catalyst, containing 531.4 eV, 532.4 eV, and 533.2 eV peaks. The vibrational peaks at 531.4 eV and 532.4 eV were respectively lattice oxygen and defect oxygen. The lattice oxygen peak area of the catalyst Ni-NiO/SiO$_2$ was larger than that of the catalyst Ni/SiO$_2$, indicating that the catalyst Ni-NiO/SiO$_2$ had more oxygen vacancies, so the catalyst had a better catalytic performance. Figure 2d shows two fitted peaks at binding energies of 102.6 eV and 103.5 eV for Si 2p in the high-resolution spectrum.

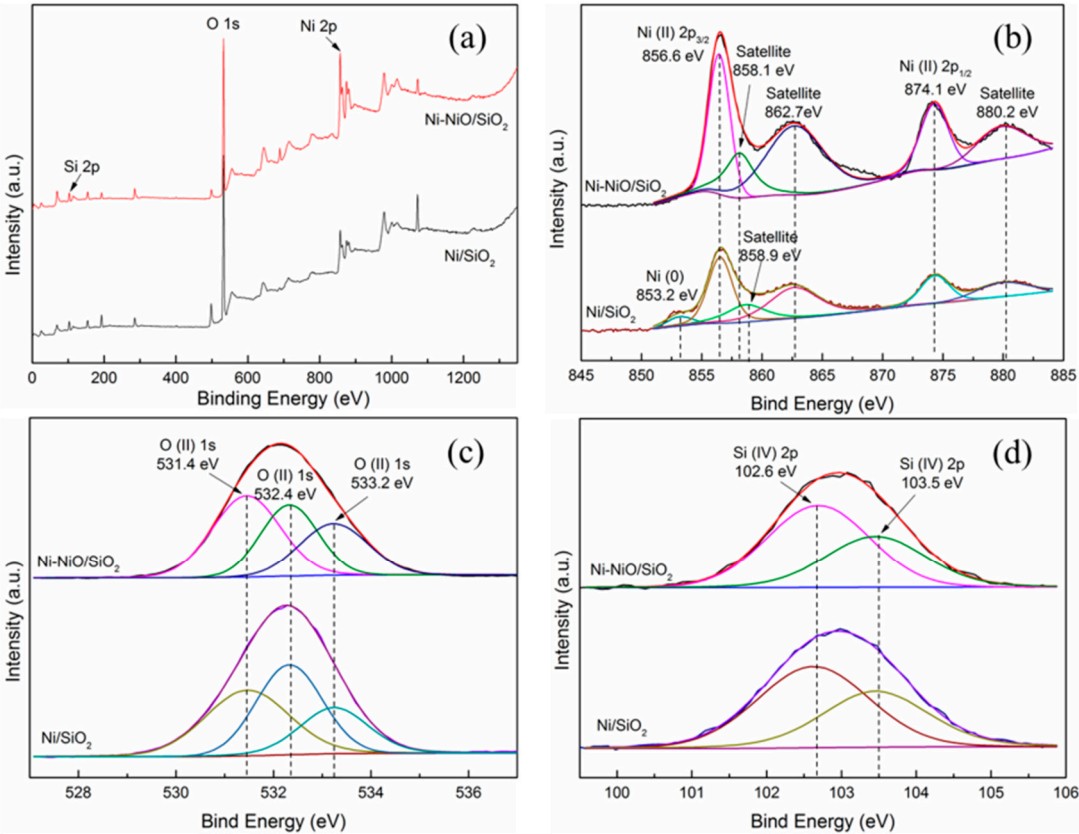

**Figure 2.** XPS spectrum of catalysts Ni-NiO/SiO$_2$ and Ni/SiO$_2$: (**a**) survey, (**b**) Ni 2p, (**c**) O 1s, and (**d**) Si 2p.

Figure 3 shows transmission electron microscopy (TEM) and high resolution TEM (HRTEM) images of the Ni-NiO/SiO$_2$ and Ni/SiO$_2$ catalysts. Figure 3a shows a TEM image of Ni-NiO/SiO$_2$ catalysts. It can be seen that the catalyst had a veil-like structure, which indicates better Ni dispersion. As shown in Figure 3b, there were two lattice fringes of d = 0.241 and 0.203 nm in the catalysts of Ni-NiO/SiO$_2$, which correspond to the (111) plane of the NiO phase and the (111) plane of the Ni

phase, respectively. The TEM image of Ni/SiO$_2$ catalysts after calcination and reduction is shown in Figure 3c, and Figure 3d reveals the HRTEM figure of Ni/SiO$_2$ catalysts, which contains a lattice fringe of d = 0.203 nm meeting the (111) crystal plane of Ni. Figure 3e–f shows the nitrogen adsorption and desorption curves and pore volume distributions of the two catalysts. Both catalysts exhibited type IV isothermal adsorption, suggesting that they were both mesoporous. Structural data for the catalyst is shown in Table 1. The specific surface area of the Ni-NiO/SiO$_2$ and Ni/SiO$_2$ catalysts was 71 and 25 m$^2$g$^{-1}$, and the average pore diameter was 16.7 and 28.4 nm, respectively. The metal dispersion of the two catalysts was tested by hydrogen pulse adsorption. The dispersion of the Ni-NiO/SiO$_2$ catalysts was 0.16%, and the dispersion of the Ni/SiO$_2$ catalysts was 0.05%, indicating that the Ni-NiO/SiO$_2$ catalysts exhibited better dispersion. The Brunauer–Emmett–Teller (BET) characterization results and the hydrogen pulse adsorption test results showed that the catalyst of Ni-NiO/SiO$_2$ had a larger specific surface area and better dispersibility, and thus had a better performance.

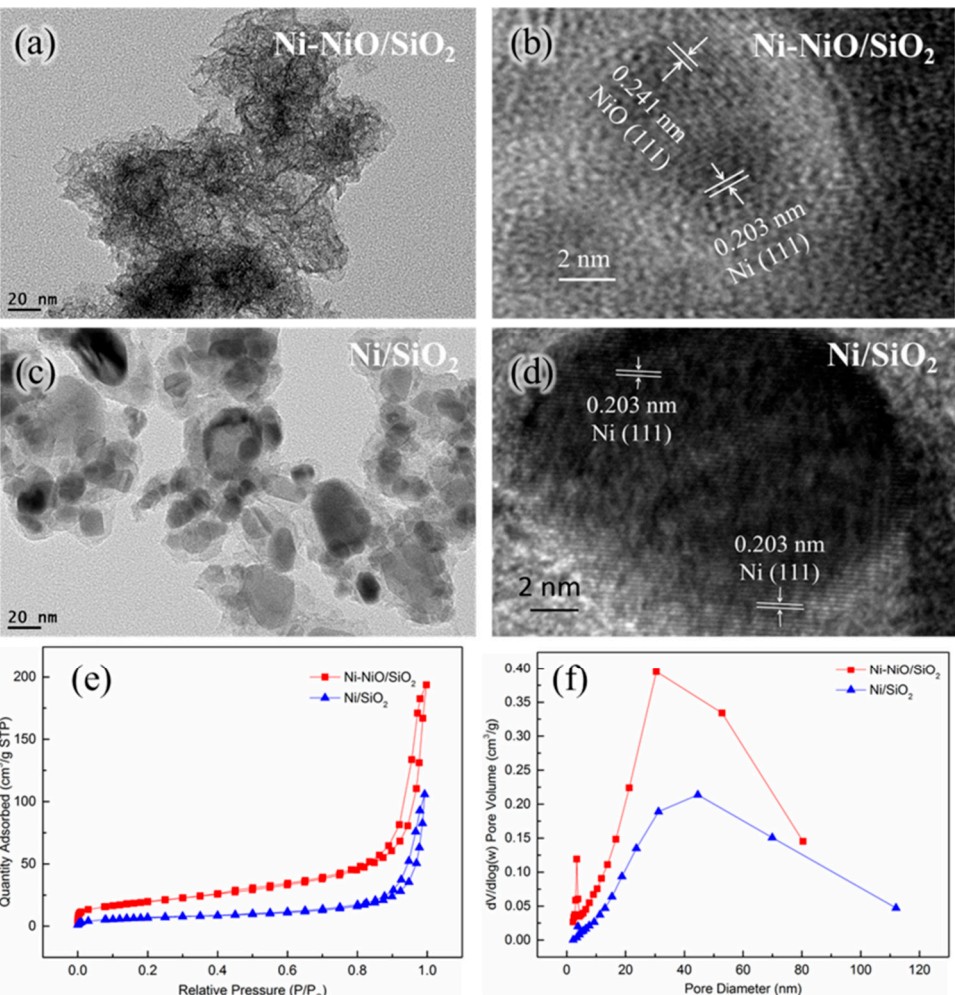

**Figure 3.** TEM and HRTEM images of (**a**,**b**) Ni-NiO/SiO$_2$ and (**c**,**d**) Ni/SiO$_2$. (**e**) N$_2$ adsorption–desorption isotherms and (**f**) pore size distribution for Ni-NiO/SiO$_2$ and Ni/SiO$_2$, respectively.

**Table 1.** Specific surface area, pore size, and pore volume of the Ni-NiO/SiO$_2$ and Ni/SiO$_2$ catalysts.

| Samples | S$_{BET}$ (m$^2$g$^{-1}$) [a] | Pore Volume (cm$^3$g$^{-1}$) [b] | Pore Size (nm) [b] |
|---|---|---|---|
| Ni-NiO/SiO$_2$ | 71 | 0.28 | 16.7 |
| Ni/SiO$_2$ | 25 | 0.16 | 28.4 |

[a] Obtained from the BET method. [b] Obtained from the Barett-Joyer-Halenda (BJH) desorption average pore volume and pore diameter.

The reduction capacity of the catalyst surface is crucial for selective catalytic reduction. Therefore, we conducted a hydrogen temperature-programmed reduction ($H_2$-TPR) test on the catalyst, and the corresponding results are shown in Figure 4. The dispersion of the NiO peak position and the mutual interaction with the carrier were assessed. The reduction peak of the Ni-NiO/SiO$_2$ catalysts at 152.0 °C was the surface NiO, while the reduction peak at 593.9 °C was the NiO interacting with the carrier [24]. The catalysts of NiO/SiO$_2$ showed reduction peaks at 609.4 °C and 765.1 °C. The difference in the reduction peak positions of the two catalysts may be due to the high NiO dispersion in the catalysts of Ni-NiO/SiO$_2$ allowing some of the NiO to be reduced to Ni at a lower temperature. The reduction peak of NiO/SiO$_2$ catalysts shifted to higher temperatures, indicating that NiO interacts strongly with the SiO$_2$ carrier. The hydrogen consumption of the two catalysts was calculated. The hydrogen consumption of the Ni-NiO/SiO$_2$ and NiO/SiO$_2$ catalysts was 44.3 cc and 86.6 cc. The difference in hydrogen consumption reflects the difference in the interaction between the SiO$_2$ and NiO. Both catalysts had two reduction peaks at different positions and different peak areas indicating different hydrogen consumption, corresponding to different reduction rates [25]. For the catalyst of Ni-NiO/SiO$_2$, the reduction rate of the first reduction temperature was 3.6%, and the reduction rate of the second reduction temperature was 96.4%. The reduction rates were 98.8% and 1.2% of the NiO/ SiO$_2$ catalysts for the two reduction temperatures.

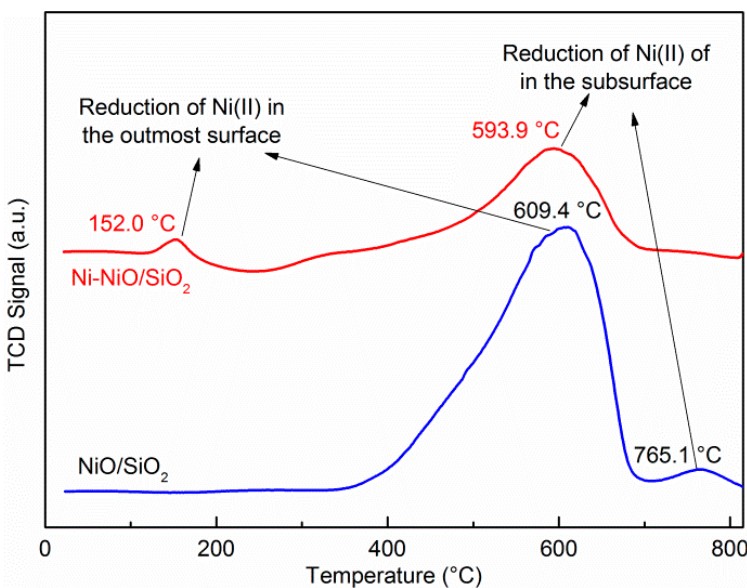

**Figure 4.** $H_2$-TPR patterns of Ni-NiO/SiO$_2$ and Ni/SiO$_2$.

Figure 5 shows a catalyst performance chart in the range 250 °C–550 °C, where the test conditions were 0.1 MPa and the space velocity was 19, 500 mL·g$^{-1}$·h$^{-1}$. As is shown in Figure 5a, the CO conversion of the Ni-NiO/SiO$_2$ and Ni/SiO$_2$ catalysts increased with the reaction temperature in the range 250 °C–300 °C and remained mainly stable in the range 300 °C–400 °C. The main reason for the decrease in catalyst performance after 400 °C was the fact that the high temperature made the catalyst agglomerate and caused carbon deposition during the reaction. These two phenomena were the main reasons for the catalyst deactivation during CO methanation. The CO conversion of the Ni-NiO/SiO$_2$ and Ni/SiO$_2$ catalysts was 97.1% and 88.7%, which were the highest at 350 °C. The corresponding CH$_4$ selectivity of the two catalysts was 81.9% and 75.3%, respectively. The turnover frequency (TOF) of the Ni-NiO/SiO$_2$ catalysts was 2.3 s$^{-1}$, while that of Ni/SiO$_2$ catalysts was 7.1 s$^{-1}$ at 350 °C. Compared with Ni-NiO/SiO$_2$ and Ni/SiO$_2$ catalysts, the performance and stability of NiO/SiO$_2$ catalysts were poor. Therefore, only Ni-NiO/SiO$_2$ and Ni/SiO$_2$ catalysts were systematically studied.

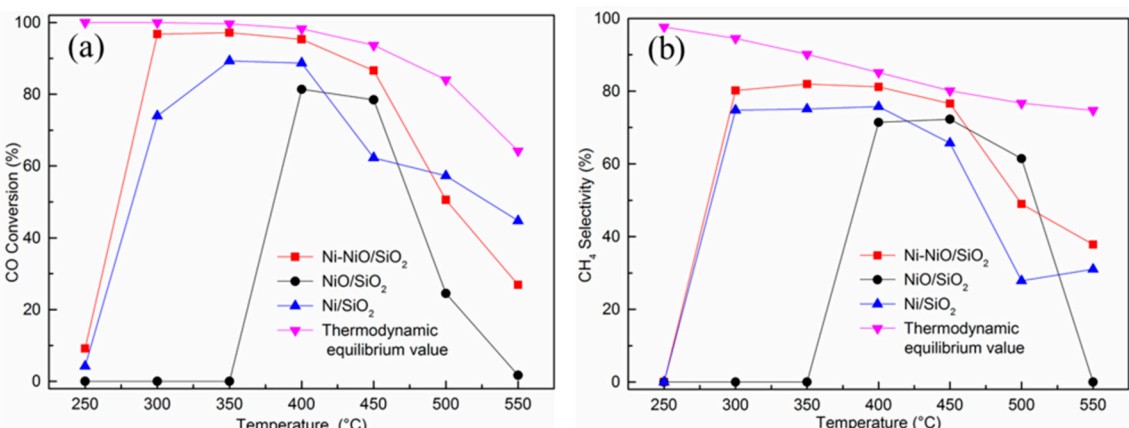

**Figure 5.** (**a**) CO conversion and (**b**) CH$_4$ selectivity of as-obtained Ni-NiO/SiO$_2$, NiO/SiO$_2$, and Ni/SiO$_2$.

In order to study the structure of the catalyst after the activity test, the two catalysts were characterized. The two catalysts after the activity test were named Ni-NiO/SiO$_2$-550 and Ni/SiO$_2$-550, respectively. The XRD pattern of the catalyst after the 250 °C–550 °C activity test is shown in Figure 6. The two catalysts had Ni diffraction peaks at 2θ = 44.5°, 51.8°, and 76.4° (PDF-# NO. 44-0850), and the three diffraction peaks correspond to the (111), (200), and (220) phases of Ni, respectively. Compared with the XRD pattern of the pre-reaction catalyst, the two catalysts had a C diffraction peak at 2θ = 26.6° (PDF-# NO. 26-1076), indicating that there was C formation in the catalyst after the reaction, which could be an important cause of catalyst inactivation. The C peak for Ni/SiO$_2$-550 catalysts was stronger than that for Ni-NiO/SiO$_2$-550, indicating that carbon deposition was more severe. This showed that Ni-NiO/SiO$_2$-550 could improve the resistance to carbon deposition [26].

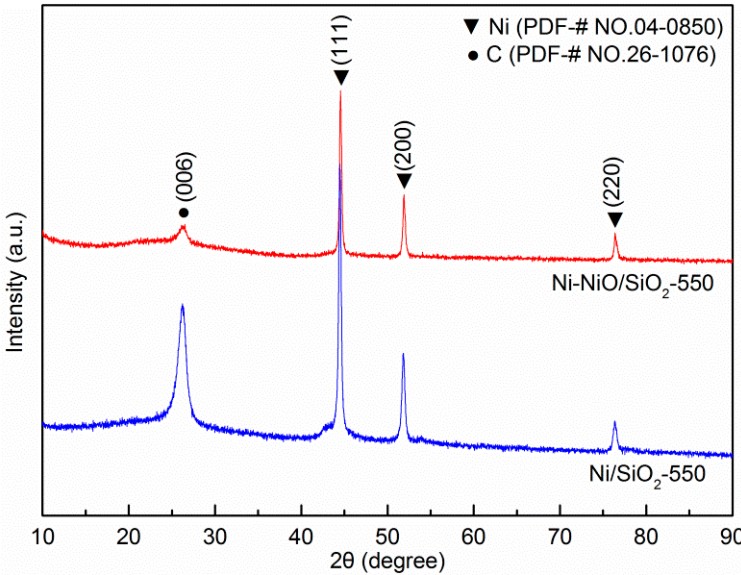

**Figure 6.** XRD patterns of the Ni-NiO/SiO$_2$-550 and Ni/SiO$_2$-550 catalysts.

TEM and HRTEM images of the catalysts after the activity test are shown in Figure 7. From Figure 7a,c, it is clear that both of the catalysts generated a large number of carbon nanotubes after the reaction, which is a critical factor of catalyst deactivation. The HRTEM images in Figure 7b,d clearly show the lattice fringes on the two catalysts. Each catalyst had two lattice fringes of d = 0.335 and 0.203 nm, which correspond to the (006) crystal plane of C and (111) crystal plane of Ni. Furthermore, Ni/SiO$_2$-550 produced more carbon nanotubes than Ni-NiO/SiO$_2$-550, indicating more serious carbon deposition, which was consistent with the results of the XRD analysis. From Figure 7a,c, we can

see that the Ni particles in Ni/SiO$_2$-550 were larger than those in Ni-NiO/SiO$_2$-550, suggesting that agglomeration occurred during the reaction, leading to catalyst deactivation.

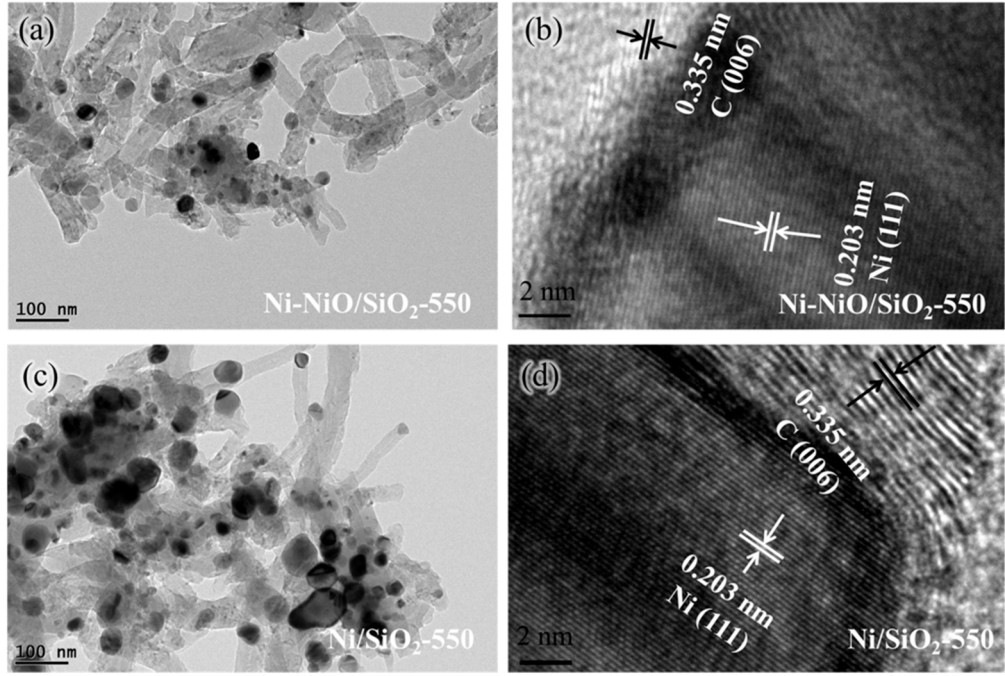

**Figure 7.** TEM and HRTEM images of (**a,b**) Ni-NiO/SiO$_2$-550 and (**c,d**) Ni/SiO$_2$-550.

In the process of CO methanation, carbon formation and deposition is one of the main causes of catalyst deactivation. Thermal gravimetric analysis (TGA) is a common means for characterizing carbon deposition in catalysts [27]. Therefore, the thermogravimetric method was employed to analyze the catalyst after the reaction. The results of the study are shown in Figure 8. The catalysts of Ni-NiO/SiO$_2$ and Ni/SiO$_2$ showed weight gain in the range of 420 to 500 °C and 420 to 530 °C, respectively, possibly due to the oxidation of Ni nanoparticles to NiO. By comparing the weight loss of the two catalysts, it was found that the carbon content of the Ni-NiO/SiO$_2$ catalyst was 15.86%, while the Ni/SiO$_2$ catalyst's carbon content was 18.74%. The removal temperatures of carbon in the two catalysts were different; the catalyst of Ni-NiO/SiO$_2$ was 500–790 °C and Ni/SiO$_2$ was 530–690 °C. They were mainly deactivating carbon species deriving from CH$_4$ decomposition, which cannot be removed until a high temperature. [24,28]. The weight of the Ni/SiO$_2$ catalysts was increased after 690 °C, possibly because the unreacted Ni was exposed and oxidized after carbon deposition was removed. Combined with XRD and TEM characterization results, it was found that the catalysts of Ni-NiO/SiO$_2$ had a better carbon deposition resistance and activity.

To test the catalyst stability, the two catalysts were tested at 350 °C for 50 h, and the results are shown in Figure 9, which indicates the change in the CO conversion of the two catalysts within 50 h. The CO conversion of Ni-NiO/SiO$_2$ catalysts decreased from 97.8% to 97.2%, while that of Ni/SiO$_2$ catalysts decreased from 78.9% to 72.8%. The inside figure shows the change in the CH$_4$ selectivity of the catalyst. It was clear that the CH$_4$ selectivity of the two catalysts remained mainly constant within 50 h at 350 °C. The test results showed that the catalytic performance and stability of Ni-NiO/SiO$_2$ catalysts were better than those of Ni/SiO$_2$.

Figure 10 shows the XRD pattern of the catalyst after 50-h testing at 350 °C. The two catalysts after the stability test were named Ni-NiO/SiO$_2$-350 and Ni/SiO$_2$-350, respectively. Both catalysts showed Ni diffraction peaks at 2θ = 44.5°, 51.8°, and 76.3° (PDF-# NO. 04-0850). However, their peak intensities were significantly different. The XRD peaks of Ni of Ni-NiO/SiO$_2$-350 were more intense than those of Ni/SiO$_2$-350. The grain size of Ni in the catalyst after the stability test was calculated by

Scherrer's formula. The grain sizes of Ni(111) of the catalysts Ni-NiO/SiO$_2$ and Ni/SiO$_2$ were 16.3 nm and 20.0 nm, respectively.

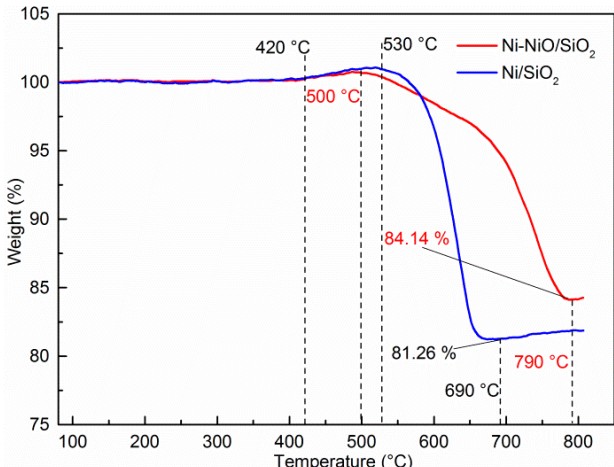

**Figure 8.** TG images of used catalysts of Ni-NiO/SiO$_2$-550 and Ni/SiO$_2$-550.

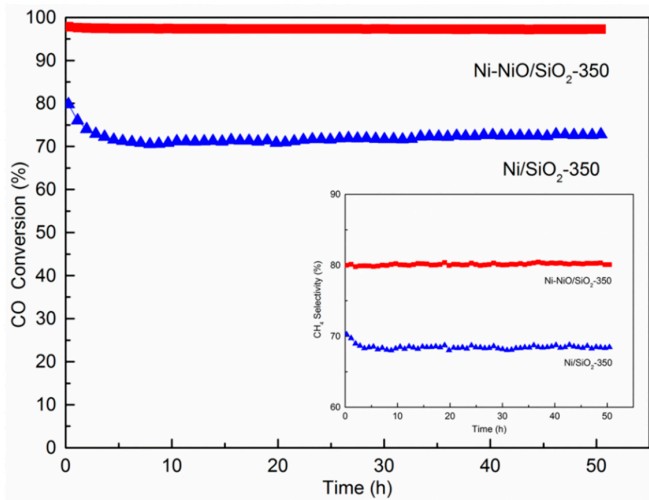

**Figure 9.** CO conversion and CH$_4$ selectivity (inside) of Ni-NiO/SiO$_2$ and Ni/SiO$_2$ catalysts.

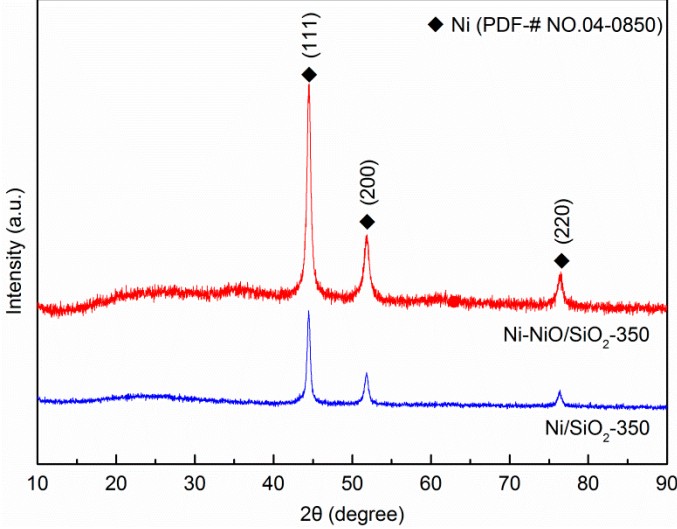

**Figure 10.** XRD patterns of catalysts Ni-NiO/SiO$_2$-350 and Ni/SiO$_2$-350.

TEM and HRTEM figures of the catalyst after 50 h testing at 350 °C are shown in Figure 11. From Figure 11a,c, it is clear that after the 50-h reaction, Ni-NiO/SiO$_2$-350 maintained its veil morphology, regardless of slight agglomeration. However, Ni/SiO$_2$-350 showed significant agglomeration. Figure 11b,d show HRTEM images of Ni-NiO/SiO$_2$-350 and Ni/SiO$_2$-350, respectively. Crystal lattices of C and Ni can be clearly seen in both pictures. After comparing these results with the XRD results before the reaction, it was found that a small quantity of C was formed on the catalyst surface after the 50-h reaction.

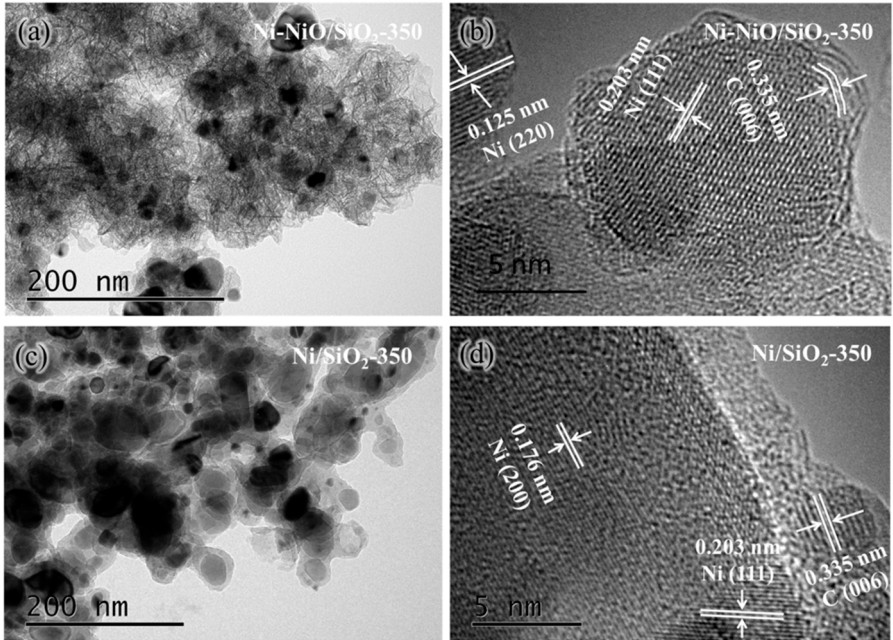

**Figure 11.** TEM and HRTEM images of the catalysts Ni-NiO/SiO$_2$ -350 (**a,b**) and Ni/SiO$_2$-350 (**c,d**).

## 3. Materials and Methods

### 3.1. Preparation of the Catalysts

The catalysts were prepared as follows. A total of 56 mL ethanol and 24 mL deionized water were put in a three-necked flask and mixed well at 30 °C. Then, Ni(NO$_3$)$_2$·6H$_2$O (1.5 g) was added to the solution and stirred it until completely dissolved. After the nickel nitrate was completely dissolved, a small amount of TEOS (0.5 mL) was used as a silicon precursor for synthesizing silica [13]. TEOS was added dropwise to the reaction vessel and stirring was continued for two minutes. Then, an NaBH$_4$ solution (0.4 g of NaBH$_4$ dissolved in 5 mL of water) was quickly added. The reaction was stopped after stirring for 1 h. After the reaction was completed, the solution was centrifuged and dried at 80 °C for 10 h. The obtained catalyst was named Ni-NiO/SiO$_2$. Some of this catalyst was calcined in air at 550 °C for 4 h and named NiO/SiO$_2$, whereas some NiO/SiO$_2$ was reduced at 500 °C in an H$_2$ atmosphere for 2 h and named Ni/SiO$_2$. The theoretical load of Ni in the catalyst of Ni-NiO/SiO$_2$ was 69.5 wt.%, while the inductive coupled plasma (ICP) characterization results showed that the load of Ni was 87.8 wt.%, which may be due to the incomplete hydrolysis of TEOS.

### 3.2. Catalyst Characterization

X-ray diffractometry was carried out to determine the crystallographic properties of the materials, and XRD patterns were obtained with a BrukerD8 Advance X-ray diffractometer (Bruker Biosciences Corporation, Billerica, MA, USA) using Cu K radiation in the 2θ range 0–90°. XPS experiments used a Thermo ESCALAB 250XIelectron spectrometer from Kratos Analytical with Mg Kα (20 mA, 12 kV) radiation (Escalab 250Xi, Thermo Fisher Scientific, Waltham, MA, USA). H$_2$-TPR experiments

were conducted to determine the reducibility of the NiO oxides by the Micromeritics TPx system (Micromeritics ASAP 2720, Micromeritics Instrument Ltd., Norcross, GA, USA) from 20 °C to 900 °C at the heat up speed of 10 °C/min using a 10 vol% $H_2$/Ar flow at 45 mL/min and maintaining the sample at 900 °C for 20 min. TEM and HRTEM images were obtained using a Tecnai G2 F20 S-TWIN(200KV)(Hillsboro, OR, USA). The BET specific surface area and BJH pore structure of the catalysts were evaluated using a Micromeritics ASAP 2020 BET apparatus (Micromeritics Instrument Ltd., Norcross, GA, USA). TGA was conducted on TG-DSC analysis system (STA 449 F3 Jupiter, Germany). ICP was used to detect the content of elements in a sample, and the sample is analyzed by inductively coupled plasma optical emission spectrometer (Agilent ICPOES730, America).

### 3.3. Activity Measurement

The catalytic performance was evaluated by a fixed-bed reactor e. The catalyst of Ni-NiO/$SiO_2$ (0.2 g) was heated to 250 °C in $N_2$ atmosphere, and the synthesis gas ($H_2$/CO = 3, the CO concentration is 23.0% mol/mol, WHSV = 19, 500 mL·$g^{-1}$·$h^{-1}$, P = 0.1 MPa) was then introduced for testing. The catalyst of Ni/$SiO_2$ (0.2 g) was reduced in an $H_2$ atmosphere for 2 h at 500 °C in the reactor and then cooled to 250 °C in an $N_2$ atmosphere. After that, synthesis gas was introduced into the reactor. The performances of the two catalysts were tested from 250 °C to 550 °C, and the effluent gas was analyzed by online gas chromatography (GC-2014C, SHIMADZU, Kyoto, Japan).

## 4. Conclusions

The catalysts of Ni-NiO/$SiO_2$ and Ni/$SiO_2$ were successfully prepared through co-precipitation with TEOS as the silicon source and Ni$(NO_3)_2$·$6H_2O$ as the nickel source. The methanation performances of the two catalysts were tested and the results showed that Ni-NiO/$SiO_2$ catalysts exhibited a good catalytic performance. The CO conversion of the Ni-NiO/$SiO_2$ catalysts reached 97.1% at 350 °C, while that of Ni/$SiO_2$ was 88.7%. The $CH_4$ selectivity of the two catalysts was 81.9% and 75.3% at 350 °C, respectively. The fresh and used catalysts were characterized through a variety of testing techniques, demonstrating that Ni-NiO/$SiO_2$ catalysts had a better dispersion, catalytic performance, and carbon deposition resistance. After testing the two catalysts at 350 °C for 50 h, it was found that the Ni-NiO/$SiO_2$ catalyst was more stable than Ni/$SiO_2$. Although the performance of the catalysts was very good, we can conduct more in-depth research to further improve the performance of the catalysts.

**Author Contributions:** F.Y. designed and administered the experiments. J.L. (Jiangwei Li) and P.L. performed experiments. Z.T. and J.L. (Jiangbing Li) collected and analyzed data. All authors discussed the data and wrote the manuscript.

**Acknowledgments:** The work was supported by the International Science and Technology Cooperation Project of Shihezi Univeristy (No. GJHZ201804), and the International Science and Technology Cooperation Project of Bingtuan (No. 2018BC002).

**Conflicts of Interest:** The authors declare no conflicts of interests.

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
