# Peer review of "Highly-Dispersed Ni-NiO Nanoparticles Anchored on an SiO2 Support for an Enhanced CO Methanation Performance"

_catalysts, doi:10.3390/catal9060506_

Round 1
Reviewer 1 Report
The article entitled Highly dispersed Ni-NiO nanoparticles anchored on SiO2 support for enhanced CO methanation performance is devoted to the preparation of three catalysts namely Ni-NiO2/SiO2, Ni/SiO2, NiO2/SiO2 and their performance in the CO methanation . Although the article is interesting the discussion and analysis of the results are poor. The comparison with other references should be improved.
Additionally the highly dispersed Ni-NiO particles should be better justified since the high concentration of the metallic phases gives a large particle size and consequently lower dispersion. The crystalline sizes of fresh catalysts are missing.
The decimal numbers in surface area results must be eliminated.
The H2 consumption and reducibility percentage from H2-TPR should be added and discussed.
The catalytic activity conditions should be better clarified e.g CO concentration and particle size, since the CO methanation performance is directly affected by both parameters.
The contribution of the reverser water gas shift to the activity results at higher temperature is missing. It should be taken into accounts since its effect is clearly visible at temperature higher than 400oC.
A discussion of the mechanism should be added since the authors must explain the effect of Ni-NiO2, Ni, NiO metallic phase on catalytic activity. Each catalyst is active in a specific windows of temperature. The authors must explain the different changes and interactions of the metallic particles with the support affecting the activity. Additionally the catalysts Ni-NiO catalysts are not calcined therefore there are others active species than can contribute to the activity with respect to the calcined and reduced catalysts.
Author Response
The article entitled Highly dispersed Ni-NiO nanoparticles anchored on SiO2 support for enhanced CO methanation performance is devoted to the preparation of three catalysts namely Ni-NiO2/SiO2, Ni/SiO2, NiO2/SiO2 and their performance in the CO methanation. Although the article is interesting the discussion and analysis of the results are poor. The comparison with other references should be improved.
1. Additionally the highly dispersed Ni-NiO particles should be better justified since the high concentration of the metallic phases gives a large particle size and consequently lower dispersion. The crystalline sizes of fresh catalysts are missing.
Reply: We thank the reviewer’s comments to our manuscript and positive evaluation of our study. We calculated the crystal size of the fresh catalyst, and the results are shown in Page 2, line 76-78: “The grain size of the fresh catalyst was calculated by the Scherrer formula [22]. The grain sizes of the catalysts Ni-NiO/SiO2, NiO/SiO2 and Ni/SiO2 were 26.8 nm, 13.6 nm and 12.6 nm, respectively.”
[22] Lv, Y.; Xin, Z.; Meng, X.; Tao, M.; Bian, Z. Ni based catalyst supported on KIT-6 silica for CO methanation: Confinement effect of three dimensional channel on NiO and Ni particles, Microporous and Mesoporous Materials 2018, 262, 89-97.
2. The decimal numbers in surface area results must be eliminated.
Reply: As suggested, we have made changes to the relevant data. Necessary modifications have been added in Page 1, line 15-17: “The as-obtained Ni-NiO/SiO2 catalyst process a high Ni content of 87.8 wt.% and exhibited a large specific surface area of 71 m2g-1 with a main pore diameter of 16.7 nm.” Page 2, line 55-56: “The catalyst Ni-Ni/SiO2 has a metal dispersion of 0.16 % and a specific surface area of 71 m2g-1, which is superior to 0.05 % and 25 m2g-1 of the catalyst Ni/SiO2.”Page 4, line 120-122: “The specific surface areas of Ni-NiO/SiO2 and Ni/SiO2 are 71 and 25 m2g−1, respectively, while the average pore diameters of Ni-NiO/SiO2 and Ni/SiO2 are 16.7 and 28.4 nm, respectively.” and we also modify the message in Page 5, Table 1, line 134-136.
Table 1. Specifical surface area, pore size and pore volume of the Ni-NiO/SiO2 and Ni/SiO2 catalysts
3. The H2 consumption and reducibility percentage from H2-TPR should be added and discussed.
Reply: As suggested, we analyzed the relevant results, the modification is included in the manuscript in Page 4, line 149-Page 5, line 157: “The hydrogen consumption of the two catalysts was calculated. The hydrogen consumption of the catalyst Ni-NiO/SiO2 was 44.3 cc, and the NiO/SiO2 was 86.6 cc. The difference in hydrogen consumption reflects the difference in the interaction between the SiO2 and NiO. Both catalysts have two reduction peaks at different positions, and different peak areas indicate different hydrogen consumption, corresponding to different reduction rates [25]. For the catalyst Ni-NiO/SiO2, the reduction rate in the first reduction temperature is 3.6 %, and the reduction Rate in the second reduction temperature is 96.4 %, and for the catalyst NiO/SiO2, in the two reduction temperatures, the reduction rates were 98.8 % and 1.2 %, respectively.
[25] Ren, J.;Li, H.; Jin, Y.; Zhu, J.; Liu, S.; Lin, J.; Li, Z. Silica/titania composite-supported Ni catalysts for CO methanation: Effects of Ti species on the activity, anti-sintering, and anti-coking properties, Applied Catalysis B: Environmental 2017, 201, 561–572.
4. The catalytic activity conditions should be better clarified e.g CO concentration and particle size, since the CO methanation performance is directly affected by both parameters.
Reply: As suggested, we describe the test conditions of the catalyst and the results are shown in Page 11, line 288-290: “The catalyst Ni-NiO/SiO2 (0.2 g) was heated to 250 °C in N2 atmosphere, and then the synthesis gas (H2/CO = 3, the CO concentration is 23.0 % mol/mol, WHSV = 19, 500 mL·g−1·h−1, P = 0.1 MPa) was introduced for testing.”
5. The contribution of the reverser water gas shift to the activity results at higher temperature is missing. It should be taken into accounts since its effect is clearly visible at temperature higher than 400°C.
Reply: The water gas reaction is a side reaction in the process of CO methanation, which has a certain influence on the catalytic performance of the catalyst. By detecting and analyzing the reaction product, we found that the CO2 content generated during the reaction is less, so we think the effect of the water gas reaction is negligible. After 400 °C, the performance of the catalyst decreased, mainly due to the agglomeration sintering of the catalyst under high temperature conditions, which is the main reason for the degradation of catalyst performance. We explain this in the text on Page 6, lines 165-168: “After 400 °C, the performance of the catalyst decreased, the main reason is that the high temperature causes the catalyst to agglomerate and the carbon deposition during the reaction. These two phenomena are the main reasons for the catalyst deactivation during the CO methanation reaction.”
6. A discussion of the mechanism should be added since the authors must explain the effect of Ni-NiO2, Ni, NiO metallic phase on catalytic activity. Each catalyst is active in a specific windows of temperature. The authors must explain the different changes and interactions of the metallic particles with the support affecting the activity. Additionally the catalysts Ni-NiO catalysts are not calcined therefore there are others active species than can contribute to the activity with respect to the calcined and reduced catalysts.
Reply: The catalytic performance of the catalysts Ni-NiO/SiO2, NiO/SiO2 and Ni/SiO2 are different. In the Ni-NiO/SiO2 catalyst, there is a synergistic effect between Ni and NiO, so it has good catalytic performance [1]. In the catalyst Ni/SiO2, Ni plays a major role in the reaction process, while the catalyst NiO/SiO2 has only NiO and no Ni exists, so its performance is poor. The three catalysts have different performance at different temperatures. The catalysts Ni-NiO/SiO2 and Ni/SiO2 have the best catalytic performance at 350 °C, and can maintain stability between 350-400 °C, while NiO/SiO2 catalytic performance is obtained at temperatures above 350 °C, and its catalytic performance is rapidly degraded after achieving the best performance at 400 °C. The best performance of the catalyst Ni-NiO/SiO2 is that the Ni-NiO particles are uniformly distributed on the surface of SiO2, and there are more active sites between Ni-NiO and the supported SiO2 [20]. Compared with Ni-NiO/SiO2, the catalyst Ni/SiO2 has a decrease in specific surface area and metal dispersion, so its active sites are reduced and its performance is degraded. Compared with the catalyst Ni/SiO2, Ni-NiO/SiO2 is not calcined and reduced, but it has good catalytic performance because there are both Ni and NiO substances in the catalyst, and there is synergy between them. It has better contact with SiO2 and has more active sites, so its performance is better.
[1] Q. Bi, X. Huang, G. Yin, T. Chen, X. Du, J. Cai, J. Xu, Z. Liu, Y. Han, F. Huang, Cooperative Catalysis of Nickel and Nickel Oxide for Efficient Reduction of CO2 to CH4, Chemcatchem 11 (2019) 1295-1302.
[20] Zhang, M.; Li, P.; Tian, Z.; Zhu, M.; Wang, F.; Li, J.; Dai, B.; Yu, F.; Qiu, H.; Gao, H. Clarification of Active Sites at Interfaces between Silica Support and Nickel Active Components for Carbon Monoxide Methanation, Catalysts 2018, 8, 293.

Reviewer 2 Report
The catalysts are active and have high stability, selectivity and activity. The reported metal dispersion is very low, below 1 %. should it be 16 and 5%. This is strange result, also thinking that the authors can see particles in XRD. Please check. Other parts are well described.
Author Response
The catalysts are active and have high stability, selectivity and activity. The reported metal dispersion is very low, below 1 %. Should it be 16 and 5%. This is strange result, also thinking that the authors can see particles in XRD. Please check. Other parts are well described.
Reply: We thank reviewer’s comment. We confirmed the metal dispersity data, the dispersion of the catalyst Ni-NiO/SiO2 was 0.16 %, and the Ni/SiO2 was 0.05 % instead of 16 % and 5 %. The dispersion of Ni in the two catalysts is low, probably due to the higher Ni content and less SiO2 content, but the dispersion of Ni on the surface of SiO2 is better, and the catalyst still has better performance.

Reviewer 3 Report
Please find attached my suggestions and comments

Author Response
In the manuscript “Highly dispersed Ni-NiO nanoparticles anchored on SiO2 support for enhanced CO methanation performance” the authors compared the catalytic activity of 87.8 wt.% Ni on SiO2 and NiO/SiO2 for CO methanation. In my opinion, this work is of general interest and the manuscript is well written. However, the manuscript is very obscure and fails to demonstrate a clear understanding of the parameters affecting catalytic behaviours due to the way of catalysts preparation. Therefore, in my opinion, the manuscript could not be accepted by Catalysts in the current version. Some issues in the manuscript are listed below:
1. Why the authors chose 87.8 wt.% Ni and how less than 25 wt. % SiO2 could act as support?
Reply: Thank you very much for the reviewer’s helpful comments to our manuscript. We chose 87.8 wt.% Ni loading because higher Ni content can provide more active sites and improve catalyst performance. The TEM characterization shows that SiO2 acts as a carrier, and Ni is uniformly distributed on the surface of SiO2, which can improve the performance of the catalyst. As described in Page 2, line 61-62: “The high dispersibility of Ni-NiO increases its activity, and higher Ni loading provides more active sites [20]”
[20] Zhang, M.; Li, P.; Tian, Z.; Zhu, M.; Wang, F.; Li, J.; Dai, B.; Yu, F.; Qiu, H.; Gao, H. Clarification of Active Sites at Interfaces between Silica Support and Nickel Active Components for Carbon Monoxide Methanation, Catalysts 2018, 8, 293.
2. What does “appropriate amount” means (line 228)?
Reply: “Appropriate amounts” is a vague description, and we have made changes to this, specific methods for catalyst preparation are described in the text, the modification is included in the manuscript in Page 10, line 258-260:“56 mL ethanol and 24 mL deionized water were placed in a three-necked flask at 30 °C and mixed well. Then add Ni(NO3)2·6H2O (1.5 g) to the solution and stir it until completely dissolved.
3. The reason for the addition of TEOS.
Reply: The reason why we add TEOS is explained in Page 10, line 260-263: “After the nickel nitrate was completely dissolved, a small amount of TEOS (0.5 mL) have been used as silicon precursors for synthesizing silica [13], and after completion of the dropwise addition, stir for two minutes, then a NaBH4 solution (0.4 g of NaBH4 dissolved in 5 mL of water) was quickly added.” TEOS is a widely used silicon source to preparation SiO2, the method of producing SiO2 with TEOS is simple, and the product has high purity and no other impurities, it is an excellent catalyst support.
[13] Lakshmanan, P.; Kim, M. S.; Park, E. D. A highly loaded Ni@SiO2 core–shell catalyst for CO methanation, Applied Catalysis A: General 2016, 513, 98-105.
4. How did the authors calculate the metal dispersion per cent (line 53)?
Reply: In order to obtain the metal dispersion, the catalyst used was characterized by H2-pulse adsorption, the amount of surface metal was calculated by the amount of hydrogen adsorbed, and then the amount of surface metal was divided by the total amount of metal to obtain a percentage of metal dispersion.
5. The authors did not explain if the heating was done on the materials in Figure 1(such as Ni-NiO/SiO2) or not.
Reply: Figure 1 shows the XRD characterization results of the catalysts Ni-NiO/SiO2, NiO/SiO2 and Ni/SiO2 after different treatments. The processing of the three catalysts is shown on Page 10, line 263-267: “After the reaction was completed, the solution was centrifuged and dried at 80 °C for 10h. The obtained catalyst was named Ni-NiO/SiO2. Some of this catalyst was calcined in air at 550 °C for 4h and named NiO/SiO2, whereas some NiO/SiO2 was reduced at 500 °C in an H2 atmosphere for 2h and named Ni/SiO2.” All three catalysts were finally cooled to room temperature, and the catalyst was not heat treated during the XRD test.
6. The authors stated (line 251): “The catalyst Ni/SiO2 (0.2 g) was reduced in a H2 atmosphere for 2 h at 500 °C in the reactor”. However, their TPR patterns (Figure 4) shows that a minimum 600°C need to reduce NiO in Ni/SiO2 material.
Reply: In the performance test, the catalyst NiO/SiO2 was reduced in a hydrogen atmosphere at 500 °C for 2 h to obtain the catalyst Ni/SiO2, in the process, only a part of NiO was reduced to Ni, not all of the catalyst was reduced. The highest point of the H2-TPR reduction peak of the catalyst NiO/SiO2 appeared at 609.4 °C, indicating that the hydrogen consumption rate was the highest at 609.4 °C. At 500 °C, the catalyst consumed a part of hydrogen, indicating that a part of the catalyst was reduced.
7. XPS spectrum of Ni/SiO2 (Figure 2b) shows characteristic peaks of NiO at around 854 eV and a satellite peak at about 862 eV. I cannot see any characteristic peaks of Ni.
Reply: As suggested, we reanalyze the XPS characterization results, as shown on Page 3, line 87-95: “Figure 2b shows the valence state of Ni in the two catalysts. The Ni 2p orbitals are in the range 850–884 eV; both catalysts contain Ni 2p3/2 and Ni 2p1/2. The Ni 2p3/2 peak can be divided into three small peaks: for catalysts Ni-NiO/SiO2, the three peaks were Ni2+ at 856.6 eV and satellite peaks of Ni 2p3/2 at 858.1 and 862.7 eV, for catalysts Ni/SiO2, the three peaks wewe at 856.6 eV, 858.9 eV and 862.7 eV. Ni 2p1/2 contains two characteristic peaks of Ni2+ at 874.1 eV and a satellite peak at 880.2 eV for both catalysts. Ni/SiO2 contains the characteristic Ni0 peak at 853.1 eV, while no peak of Ni0 was observed in the catalyst Ni-NiO/SiO2 ,the possible reason is that the elemental Ni in Ni-NiO/SiO2 is relatively dispersed and cannot be detected [9,23].”
[9] Li, P.; Wen, B.; Yu, F.; Zhu, M.; Guo, X.; Han, Y.; Kang, L.; Huang, X.; Dan, J.; Ouyang, F.; Dai, B. High efficient nickel/vermiculite catalyst prepared via microwave irradiation-assisted synthesis for carbon monoxide methanation, Fuel 2016, 171, 263-269.
[23] Li, Z.; Mo, L.; Kathiraser, Y.; Kawi, S. Yolk–Satellite–Shell Structured Ni–Yolk@Ni@SiO2 Nanocomposite: Superb Catalyst toward Methane CO2 Reforming Reaction, ACS Catalysis 2014, 4, 1526-1536.
8. TEM images in Figure 3(a) and 3(c) show that samples contain low melting point impurities.
Reply: Figures 3(a) and 3(c) are TEM images of the catalysts Ni-NiO/SiO2 and Ni/SiO2, respectively, which have no low melting point impurities. The difference in TEM images of the two catalysts is due to the fact that the catalyst Ni-NiO/SiO2 is not calcined, so SiO2 is in an amorphous state and has poor crystallinity. The catalyst Ni/SiO2 is calcined and reduced at high temperature, so it is better crystallinity.

Round 2
Reviewer 1 Report
Accept in present form
Author Response
Thank you very much for the reviewer’s comments.

Reviewer 3 Report
In general, this is a quite extensive work, and a series of characterizations have been done on the catalyst. Also, the authors have spent some effort to improve the manuscript further. However, as I mentioned before, I do not recommend the acceptance of this paper due to the way of catalysts preparation and experimental procedures. The authors used 12.2 wt% SiO2 as a carrier, which is not possible. Mole SiO2/(NiO-SiO2) less than 0.33 (equal to Ni2O4Si) cannot act as efficient support. I can not see any peaks of amorphous silica in the presented XRD patterns. I can view the melted material around particles in the TEM image in Figure 3(c), which I believe are due to local eutectic by remained precursors. I suggest the authors use a sufficient amount of SiO2 (at least 50wt%) and re-do the calcination at a higher temperature. I am sorry that I could not give a positive comment in the current paper.
Author Response
We thank the reviewer’s comments to our manuscript.
Although the weight content of active component in catalysts is widely used in catalytic reaction on account of easy calculation, actually the volume content is the key factor. Relative light catalyst supports, such as graphene [1,2], silica aerogels [3,4], etc., could support more active components and get big weight content, even more than 75 wt.%. For CO methanation catalysts, the value of Ni content in industrial catalyst is usually in the range of 30-55 wt.%. It is found that some catalysts with high Ni content (e.g., 55 wt.%, 76 wt.%, etc. [5,6]) were reported in the literatures.
As is shown in Fig. 1, the catalysts of NiO/SiO2 and Ni/SiO2 showed characteristic diffraction peaks of silica at 2θ = 23.8° (PDF-# NO.51-1380). No characteristic peak of SiO2 was detected in the Ni-NiO/SiO2 catalysts that may be attributed to the non-calcined sample, low crystal silica and large amount of Ni-NiO on the surface.
Figure 1. XRD patterns of as-obtained Ni-NiO/SiO2, NiO/SiO2 and Ni/SiO2 Catalysts (The inner picture is the diffraction peak of SiO2).
In the TEM image of Fig.3c, the material surrounding the Ni particles is SiO2, rather than the local eutectic of the remaining precursors, because the catalyst Ni/SiO2 was calcined in air at 550 °C for 4h and reduction at 500 °C for 2h in H2 atmosphere, the precursor was completely removed.
References:
[R1] M. Khan, M. Kuniyil, M. Shaik, M. Khan, S. Adil, A. Al-Warthan, H. Alkhathlan, W. Tremel, M. Tahir, M. Siddiqui, Plant Extract Mediated Eco-Friendly Synthesis of Pd@Graphene Nanocatalyst: An Efficient and Reusable Catalyst for the Suzuki-Miyaura Coupling, Catalysts 7 (2017) 20.
[R2] C. Zhao, J. Guo, Q. Yang, L. Tong, J. Zhang, J. Zhang, C. Gong, J. Zhou, Z. Zhang, Preparation of magnetic Ni@graphene nanocomposites and efficient removal organic dye under assistance of ultrasound, Applied Surface Science 357 (2015) 22-30..
[R3] Y. Zhao, Y. Liang, Q. Jia, B. Zhang, Preparation of CuO-CoO-MnO/SiO2 nanocomposite aerogels as catalyst carriers and their application in the synthesis of diphenyl carbonate, Journal of Wuhan University of Technology-Mater. Sci. Ed. 26 (2011) 595-599.
[R4] F. Shi, J. Liu, N. Tang, X. Dong, X. Zhang, L. Bai, X. Leng, Preparation of SiO2 Aerogel Supported Nano-TiO2 Photocatalysts for Removing Rhodamine B in the Waste Water, Advanced Materials Research 531 (2012) 494-498.
[R5] P. Lakshmanan, M.S. Kim, E.D. Park, A highly loaded Ni@SiO2 core–shell catalyst for CO methanation, Applied Catalysis A: General 513 (2016) 98-105.
[R6] S. Abate, K. Barbera, E. Giglio, F. Deorsola, S. Bensaid, S. Perathoner, R. Pirone, G. Centi, Synthesis, Characterization, and Activity Pattern of Ni-Al Hydrotalcite Catalysts in CO2 Methanation, Industrial & Engineering Chemistry Research 55 (2016) 8299-8308.
